# Learning Task Decomposition with Ordered Memory Policy Network

**Yuchen Lu & Yikang Shen**
University of Montreal, Mila
Montreal, Canada

**Siyuan Zhou**
Peking University
Beijing, China

**Aaron Courville**
University of Montreal, Mila, CIFAR
Montreal, Canada

**Joshua B. Tenenbaum**
MIT BCS, CBMM, CSAIL
Cambridge, United States

**Chuang Gan**
MIT-IBM Watson AI Lab
Cambridge, United States

## Abstract

Many complex real-world tasks are composed of several levels of sub-tasks. Humans leverage these hierarchical structures to accelerate the learning process and achieve better generalization. In this work, we study the inductive bias and propose Ordered Memory Policy Network (OMPN) to discover subtask hierarchy by learning from demonstration. The discovered subtask hierarchy could be used to perform task decomposition, recovering the subtask boundaries in an unstructured demonstration. Experiments on Craft and Dial demonstrate that our model can achieve higher task decomposition performance under both unsupervised and weakly supervised settings, comparing with strong baselines. OMPN can also be directly applied to partially observable environments and still achieve higher task decomposition performance. Our visualization further confirms that the subtask hierarchy can emerge in our model [1].

## 1 Introduction

Learning from Demonstration (LfD) is a popular paradigm for policy learning and has served as a warm-up stage in many successful reinforcement learning applications (Vinyals et al., 2019; Silver et al., 2016). However, beyond simply imitating the experts' behaviors, an intelligent agent's crucial capability is to decompose an expert's behavior into a set of useful skills and discover sub-tasks. The discovered structure from expert demonstrations could be leveraged to re-use previously learned skills in the face of new environments (Sutton et al., 1999; Gupta et al., 2019; Andreas et al., 2017). Since manually labeling sub-task boundaries for each demonstration video is extremely expensive and difficult to scale up, it is essential to learn task decomposition *unsupervisedly*, where the only supervision signal comes from the demonstration itself.

This question of discovering a meaningful segmentation of the demonstration trajectory is the key focus of Hierarchical Imitation Learning (Kipf et al., 2019; Shiarlis et al., 2018; Fox et al., 2017; Achiam et al., 2018) These works can be summarized as finding the optimal behavior hierarchy so that the behavior can be better predicted (Solway et al., 2014). They usually model the sub-task structure as latent variables, and the subtask identifications are extracted from a learnt posterior. In this paper, we propose a novel perspective to solve this challenge: could we design a *smarter* neural network architecture, so that the sub-task structure can emerge during imitation learning? To be specific, we want to design a recurrent policy network such that examining the memory trace at each time step could reveal the underlying subtask structure.

Drawing inspiration from the Hierarchical Abstract Machine (Parr & Russell, 1998), we propose that each subtask can be considered as a finite state machine. A hierarchy of sub-tasks can be represented as different slots inside the memory bank. At each time step, a subtask can be internally updated with the new information, call the next-level subtask, or return the control to the previous level subtask. If our designed architecture maintains a hierarchy of sub-tasks operating in the described manner,

---

[1]Project page: https://ordered-memory-rl.github.io/

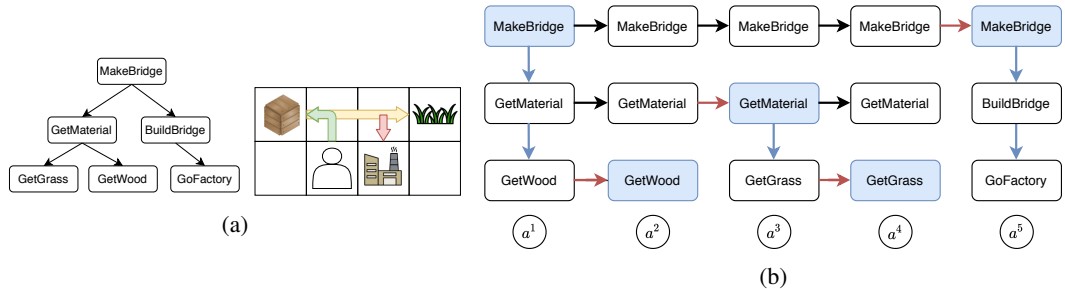

Figure 1: (a) A simple grid world with the task "make bridge", which can be decomposed into multi-level subtask structure. (b) The representation of subtask structure within the agent memory with *horizontal update* and *vertical expansion* at each time step. The black arrow indicates a copy operation. The *expansion position* is the memory slot where the vertical expansion starts and is marked blue.

then subtask identification can be as easy as monitoring when the low-level subtask returns control to the higher-level subtask, or when the high-level subtask expands to the new lower-level subtask.

We give an illustrative grid-world example in Figure 1. In this example, there are different ingredients like grass for the agent to pickup. There is also a factory where the agent can use the ingredients. Suppose the agent wants to complete the task of building a bridge. This task can be decomposed into a tree-like, multi-level structure, where the root task is divided into $GetMaterial$ and $BuildBridge$. $GetMaterial$ can be further divided into $GetGrass$ and $GetWood$. We provide a sketch on how this subtask structure should be represented inside the agent's memory during each time step. The memory would be divided into different levels, corresponding to the subtask structure. When $t = 1$, the model just starts with the root task, $MakeBridge$, and vertically expands into $GetMaterial$, which further vertically expands into $GetWood$. The *vertical expansion* corresponds to planning or calling the next level subtasks. The action is produced from the lowest-level memory. The intermediate $GetMaterial$ is copied for $t < 3$, but horizontally updated at $t = 3$, when $GetWood$ is finished. The *horizontal update* can be thought of as an internal update for each subtask, and the updated $GetMaterial$ vertically expands into a different child $GetGrass$. The root task is always copied until $GetMaterial$ is finished at $t = 4$. As a result, $MakeBridge$ goes through one horizontal update at $t = 5$ and then expands into $BuildBridge$ and $GoFactory$. We can identify the subtask boundaries from this representation by looking at the change of *expansion position*, which is defined to be the memory slot where vertical expansion happens. E.g., from $t = 2$ to $t = 3$, the expansion position goes from the lowest level to the middle level, suggesting the completion of the low-level subtask. From $t = 4$ to $t = 5$, the expansion position goes from the lowest level to the highest level, suggesting the completion of both low-level and mid-level subtasks.

Driven by this intuition, we propose the *Ordered Memory Policy Network* (OMPN) to support the subtask hierarchy described in Figure 1. We propose to use a bottom-up recurrence and a top-down recurrence to implement *horizontal update* and *vertical expansion* respectively. Our proposed memory-update rule further maintains a hierarchy among memories such that the higher-level memory can store longer-term information. At each time step, the model would softly decide the expansion position from which to perform vertical expansion based on a differentiable stick-breaking process, so that our model can be trained end-to-end. We demonstrate the effectiveness of our approach with multi-task behavior cloning. We perform experiments on both grid-world as well as more challenging robotic tasks. We show that OMPN is able to perform task decomposition in both an unsupervised and weakly supervised manner, comparing favorably with strong baselines. Meanwhile, OMPN still maintains the similar, if not better, performance on behavior cloning in terms of sample complexity and returns. Our ablation study shows the contribution of each component in our architecture. Our visualization further confirms that the subtask hierarchy emerges in our model's expanding positions.

## 2    ORDERED MEMORY POLICY NETWORK

We describe our policy architecture given the intuition described above. Our model is a recurrent policy network $p(a^t|s^t, M^t)$ where $M \in \mathcal{R}^{n \times m}$ is a block of $n$ memory while each memory has

dimension $m$. We use $M_i$ to refer to the $i$th slot of the memory, so $M = [M_1, M_2, ..., M_n]$. The highest-level memory is $M_n$ while the lowest-level memory is $M_1$. Each memory can be thought of as the representation of a subtask. We use the superscript to denote the time step $t$.

At each time step, our model will first transform the observation $s^t \in \mathcal{S}$ to $x^t \in \mathcal{R}^m$. This can be achieved by a domain-specific observation encoder. Then we have an ordered-memory module $M^t, O^t = OM(x^t, M^{t-1})$ to generate the next memory and the output. The output $O^t$ is sent into a feed-forward neural net to generate the action distribution.

## 2.1 ORDERED MEMORY MODULE

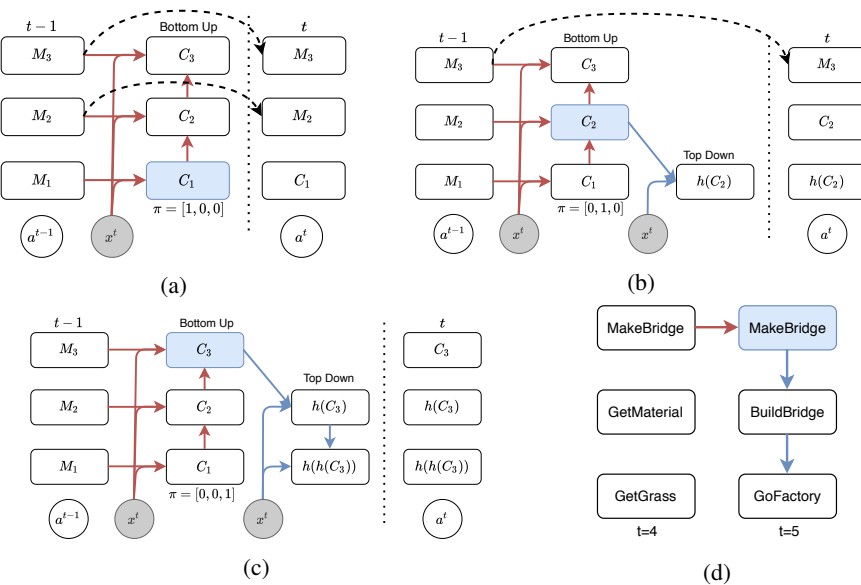

Figure 2: Dataflow of how $M^{t-1}$ will be updated in $M^t$ for three memory slots when the expansion position is at a (a) low, (b) middle, or (c) high position. Blue arrows and red arrows corresponding to the vertical expansions and horizontal updates. (d) is a snapshot of $t = 5$ from the grid-world example Figure 1b. The subtask-update behavior corresponds to the memory-update when the expansion position is at the high position.

The ordered memory module first goes through a bottom-up recurrence. This operation implements the *horizontal update* and updates each memory with the new observation. We define $C^t$ to be the updated memory:

$$C_i^t = \mathcal{F}(C_{i-1}^t, x^t, M_i^{t-1})$$

for $i = 1, ..., n$ where $C_0^t = x^t$ and $\mathcal{F}$ is a cell function. Different from our mental diagram, we make it an recurrent process since the high-level memory might be able to get information from the updated lower-level memory in addition to the observation. In our experiment we find that such recurrence will help model perform better than in task decomposition. For each memory, we also generate a score $f_i^t$ from 0 to 1 with $f_i^t = \mathcal{G}(x^t, C_i^t, M_i^t)$ for $i = 1, ..., n$. The score $f_i^t$ can be interpreted as the probability that subtask $i$ is completed at time $t$.

In order to properly generate the final *expansion position*, we would like to insert the inductive bias that the higher-level subtask is expanded only if the higher-level subtask is not completed while all the lower-level subtasks are completed, as is shown in Figure 1b. As a result we use a stick-breaking process as follows:

$$\hat{\pi}_i^t = \begin{cases} (1 - f_i^t) \prod_{j=1}^{i-1} f_j^t & 1 < i \le n \\ 1 - f_1^t & i = 1 \end{cases}$$

Finally we have the expansion position $\pi_i^t = \hat{\pi}_i^t / \sum \hat{\pi}^t$ as a properly normalized distribution over $n$ memories. We can also define the ending probability as the probability that every subtask is finished.

$$\pi_{end}^t = \prod_{i=1}^{n} f_i^t \tag{1}$$

Then we use a top-down recurrence on the memory to implement the vertical expansion. Starting from $\hat{M}_n^t = 0$, we have

$$\hat{M}_i^t = h(\bar{\pi}_{i+1}^t C_{i+1}^t + (1 - \bar{\pi}_{i+1}^t)\hat{M}_{i+1}^t, x^t),$$

where $\vec{\pi}_i^t = \sum_{j \geq i} \pi_j^t$, $\bar{\pi}_i^t = \sum_{j \leq i} \pi_j^t$, and $h$ can be any cell function. Then we update the memory in the following way:

$$M^t = M^{t-1}(1 - \vec{\pi}^t) + C^t \pi^t + \hat{M}^t(1 - \bar{\pi}^t) \tag{2}$$

where the output is read from the lowest-level memory $O^t = M_1^t$. For better understanding purpose, we show in Figure 2 how $M^{t-1}$ will be updated into $M^t$ with $n = 3$, when the expansion position is at a high, middle and low position respectively. The memory higher than the expansion position will be preserved, while the memory at and lower than the expansion position will be over-written. We also take the snapshot of $t = 5$ from our the early example in Figure 1b and show that the subtask-update behavior corresponds to our memory-update when the expansion position is at the high position.

Although we show only the discrete case for illustration, the vector $\pi^t$ is actually continuous. As a result, the whole process is fully differentiable and can be trained end-to-end. More details can be found in the appendix A.

The memory depths $n$ is a hyper-parameter here. If $n$ is too small, we might not have enough capacity to cover the underlying structure in the data, If $n$ is too large, we might impose some extra optimization difficulty. In our experiments we investigate the effect of using different $n$.

## 2.2 UNSUPERVISED TASK DECOMPOSITION WITH BEHAVIOR CLONING

We assume the following setting. We firstly have a *training phase* to perform behavior cloning on an unstructured demonstration dataset with state-action pairs. Then during the *detection phase*, the user would specify the number of subtasks $K$ for the model to produce the task boundaries.

We firstly describe our *training phase*. We develop a unique regularization technique which can help our model learning the underlying hierarchy structure. Suppose we have an action space $\mathcal{A}$. We first augment this action space with $\mathcal{A}' = \mathcal{A} \cup \{done\}$, where $done$ is a special action. Then we can modify the action distribution accordingly:

$$p'(a^t|s^t) = \begin{cases} p(a^t|s^t)(1 - \pi_{end}^t) & a^t \in \mathcal{A} \\ \pi_{end}^t & a^t = done \end{cases}$$

Then for each demonstration trajectory $\tau = \{s^t, a^t\}_{t=1}^T$, we transformed it into $\tau' = \tau \cup \{s^{T+1}, a^{T+1} = done\}$, which is essentially telling the model to output $done$ only after the end of the trajectory. This process can be achieved on both discrete and continuous action space without heavy human involvement described in Appendix A. Then we will maximize $\sum_{t=1}^{T+1} \log p'(a^t|s^t)$ on $\tau'$. We find that including $\pi_{end}^t$ into the loss is crucial to prevent our model degenerating into only using the lowest-level memory, since it provides the signal to raise the expansion position at the end of the trajectory, benefiting the task decomposition performance. We also justify this in our ablation study.

Since the expansion position should be high if the low-level subtasks are completed, we can achieve unsupervised task decomposition by monitoring the behavior of $\pi^t$. To be specific, we define $\pi_{avg}^t = \sum_{i=1}^n i\pi_i^t$ as the expected expansion position. Given $\pi_{avg}$, we consider the following methods to recover the subtask boundaries.

**Top-K** In this method we choose the time steps of $K$ largest $\pi_{avg}$ to detect the boundary, where $K$ is the desired number of sub-tasks given by the user during the detection phase. We find that this method is suitable for the discrete action space, where there is a very clear boundary between subtasks.

**Thresholding** In this method we standardize the $\pi_{avg}$ into $\hat{\pi}_{avg}$ from 0 to 1, and then we compute a Boolean array $\mathbb{1}(\pi_{avg} > thres)$, where $thres$ is from 0 to 1. We retrieve the subtask boundaries from the ending time step of each $True$ segments. We find this method is suitable for continuous control settings, where the subtask boundaries are more ambiguous and smoothed out across time steps. We also design an algorithm to automatically select the threshold in Appendix B.

## 3 RELATED WORK

Our work is related to option discovery and hierarchical imitation learning. The existing option discovery works have focused on building a probabilistic graphical model on the trajectory, with options as latent variables. DDO (Fox et al., 2017) proposes an iterative EM-like algorithm to discover multiple level of options from the demonstration. DDO was later applied in the continuous action space (Krishnan et al., 2017) and program modelling (Fox et al., 2018). Recent works like compILE (Kipf et al., 2019) and VALOR (Achiam et al., 2018) also extend this idea by incorporating more powerful inference methods like VAE (Kingma & Welling, 2013). Lee (2020) also explore unsupervise task decompostion via imitation, but their method is not fully end-to-end, requires an auxiliary self-supervision loss, and does not support multi-level structure. Our work focuses on the role of neural network inductive bias in discovering re-usable options or subtasks from demonstration. We do not have an explicit "inference" stage in our training algorithm to infer the option/task ID from the observations. Instead, this inference "stage" is implicitly designed into our model architecture via the stick-breaking process and expansion position. Based on these considerations, we choose compILE as the representative baseline for this field of work.

Our work is also related to Hierarchical RL (Vezhnevets et al., 2017; Nachum et al., 2018; Bacon et al., 2017). These works usually propose an architecture that has a high-level controller to output a goal, while the low-level architecture takes the goal and outputs the primitive actions. However, these works mainly deal with the control problem, and do not focus on learning task decomposition from the demonstration. Recent works (Gupta et al., 2019; Lynch et al., 2020) also include hierarchical imitation learning stage as a way to pretraining low-level policies before apply Hierarchical RL for finetuning, however they do not produce the task boundaries and therefore are not comparable to our works. Moreover, their hierarchical IL algorithm exploits the fact that the goal can be described as a point in the state space, so that they are able to re-label the ending state of an unstructured demonstrations as a fake goal to train the goal-conditioned policy. Meanwhile our approach is designed to be general. In addition to the option framework, our work is closely related to Hierarchical Abstract Machine (HAM) (El Hihi & Bengio, 1996). Our concept of subtask is similar to the finite state machine (FSM). The horizontal update corresponds to the internal update of the FSM, while the vertical expansion corresponds to calling the next level of the FSM. Our stick-breaking process is also a continuous realization of the idea that low-level FSM transfers control back to high-level FSM at completion.

Recent work (Andreas et al., 2017) introduces the modular policy networks for reinforcement learning so that it can be used to decompose a complex task into several simple subtasks. In this setting, the agent is provided a sequence of subtasks, called *sketch*, at the beginning. Shiarlis et al. (2018) propose TACO to jointly learn sketch alignment with action sequence, as well as imitating the trajectory. This work can only be applied in the "weakly supervised" setting, where they have some information like the sub-task sequence. Nevertheless, we also choose TACO (Shiarlis et al., 2018) as one of our baselines.

Incorporating varying time-scale for each neuron to capture hierarchy is not a new idea (Chung et al., 2016; El Hihi & Bengio, 1996; Koutnik et al., 2014). However, these works do not focus on recovering the structure after training, which makes these methods less interpretable. Shen et al. (2018) introduce Ordered Neurons and show that they can induce syntactic structure by examining the hidden states after language modelling. However ONLSTM does not provide mechanism to achieve the top-down and bottom-up recurrence. Our model is mainly inspired by the Ordered Memory (Shen et al., 2019). However, unlike previous work our model is a decoder expanding from root task to subtasks, while the Ordered Memory is an encoder composing constituents into sentences. Recently Mittal et al. (2020) propose to combine top-down and bottom-up process. However their main motivation is to handle uncertainty in the sequential prediction and they do not maintain a hierarchy of memories with different update frequencies.

## 4 EXPERIMENT

We would like to evaluate whether OMPN is able to jointly learning task decomposition during behavior cloning. We would like to answer the following questions in our experiments.

**Q1:** Can OMPN be applied in both continuous and discrete action space?

**Q2:** Can OMPN be applied in both unsupervised, as well as, weakly supervised setting for task decomposition?

**Q3:** How much does each component helps the task decomposition?

**Q4:** How does the task decomposition performance change with different hyper-parameters, e.g., memory dimension $m$ and memory depths $n$?

## 4.1 SETUP AND METRICS

For the discrete action space, we use a grid world environment called Craft adapted from Andreas et al. (2017)[2]. At the beginning of each episode, an agent is equipped with a task along with the sketch, e.g. $makecloth = (getgrass, gofactory)$. The original environment is fully observable. To further test our model, we make it also support partial observation by providing a self-centric window. For the continuous action space, we have a robotic setting called Dial (Shiarlis et al., 2018) where a JACO 6DoF manipulator interact with a large number pad[3]. For each episode, the sketch is a sequence of numbers to be pressed. More details on the demonstration can be found in Appendix E.

We experiment in both unsupervised and weakly supervised settings. For the unsupervised setting, we did not provide any task information. For the weakly supervised setting, we provide the subtask sequence, or sketch, in addition to the observation. We use *nosketch* and *sketch* to denote these two settings respectively. We choose compILE to be our unsupervised baseline while TACO to be our weakly supervised baseline.

We use the ground-truth $K$ to get the best performance of both our models and the baselines. This setting is consistent with the previous literature (Kipf et al., 2019). The details about task decomposition metric can be found in the appendix C.

## 4.2 TASK DECOMPOSITION RESULTS

| | | Craft | | | | Dial |
|---|---|---|---|---|---|---|
| | | Full | | Partial | | |
| | | Align Acc | F1(tol=1) | Align Acc | F1(tol=1) | Align Acc |
| NoSketch | OMPN | **93(1.7)** | 95(1.2) | **84(6)** | **89(4.6)** | **87(4.0)** |
| | compILE | 86(1.4) | **97(0.8)** | 54(1.4) | 57(4.8) | 45(5.2) |
| Sketch | OMPN | **97(1.2)** | 98(0.9) | **72(6.2)** | 83(8.1) | 82(7.4) |
| | TACO | 90(3.6) | - | 66(2.2) | - | **98(0.1)** |

Table 1: Alignment accuracy and F1 scores with tolerance 1 on Craft and Dial. The results are averaged over five runs, and the number in the parenthesis is the standard deviation.

In this section, we mainly address the question **Q1** and **Q2**. Our main results for task decomposition in Craft and Dial are in Table 1. In Craft, we use the $TopK$ detection algorithm. Our results show that OMPN is able to outperform baselines in both unsupervised and weakly-supervised settings with a higher F1 scores and alignment accuracy. In general, there is a decrease of the performance for all models when moving from full observations to partial observations. Nevertheless, compared with the baselines, we find that OMPN suffers less performance drop than the baselines. The F1 score results with different $K$ other than the ground truth is in Table 5 and Table 6.

In Dial, we use the automatic threshold method described in Appendix B. Our results is able to outperform compILE for the unsupervised setting, but is not better than TACO when the sketch information is given. In general, our current model does not benefit from the additional sketch information, and we hypothesize that it is because we use a very trivial way of incorporating sketch information by simple concatenating it with the observation. A more effective way would be using attention over the sketch sequence to properly feed the related task information into the model. The F1 score results with different $K$ other than the ground truth is in Table 8.

---

[2]https://github.com/jacobandreas/psketch
[3]https://github.com/KyriacosShiarli/taco/tree/master/taco/jac

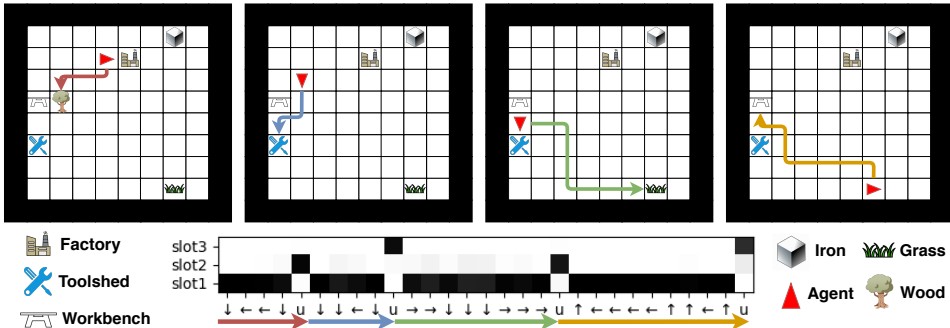

Figure 3: Visualization of the learnt expanding positions on Craft. We present $\pi$ and the action sequence. The four subtasks are $GetWood$, $GoToolshded$, $GetGrass$ and $GoWorkbench$. In the action sequence, "u" is either picking up/using the object. Each ground truth subtask is highlighted with an arrow of different colors.

### 4.3  QUALITATIVE ANALYSIS

In this section, we provide some visualization of the learnt expanding positions to qualitatively evaluate whether OMPN can operate with a subtask hierarchy.

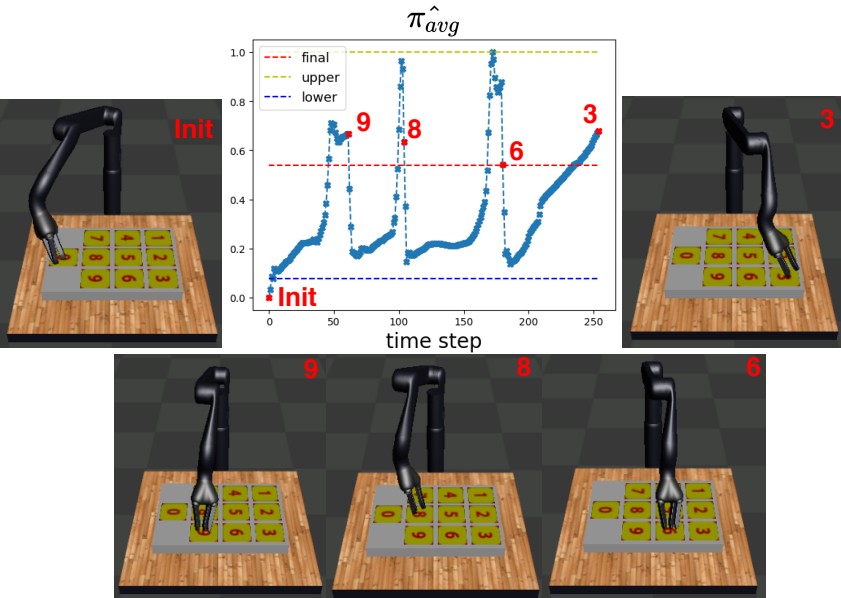

Figure 4: Visualization of the learnt expanding positions of Dial domain. The task is to press the number sequence $[9, 8, 6, 3]$. We plot the $\hat{\pi}_{avg}$. Our threshold selection algorithm produce a upper bound and and lower bound, and the final threshold is computed as the average. The task boundary is detected as the last time step of a segment above the final threshold. The frames at the detected task boundary show that the robot just finishes each subtask.

We firstly visualize the behavior of expanding positions for Craft in Figure 3. We find that at the end of each subtask, the model learns to switch to a higher expansion position when facing the target object/place, while within each subtask, the model learns to maintain the lower expansion position. This is our desired behavior described in Figure 1. What is interesting is that although the ground truth hierarchy structure might be only two-levels, our model is able to learn multiple level hierarchy if given some redundant depths. In this example, the model combines the first two subtasks into one group and the last two subtasks into another group. More results can be found in Appendix F.

In Figure 4, we show the qualitative result in Dial. We find that, instead of having a sharp subtask boundary, the boundary between subtasks is more ambiguous with high expanding positions

across multiple time steps, and this motivates our design of the threshold algorithm. This happens because we use the observation to determine the expanding position. In Dial, the state difference are less significant due to a small time skip and continuous space, so the expanding position near the boundaries might be high for multiple time steps. Also just like in Craft, the number of subtasks naturally emerge from the peak patterns. We also provide addition qualitative result on Kitchen released by Gupta et al. (2019) in Figure 5. More results on Kitchen can be found in Appendix I.

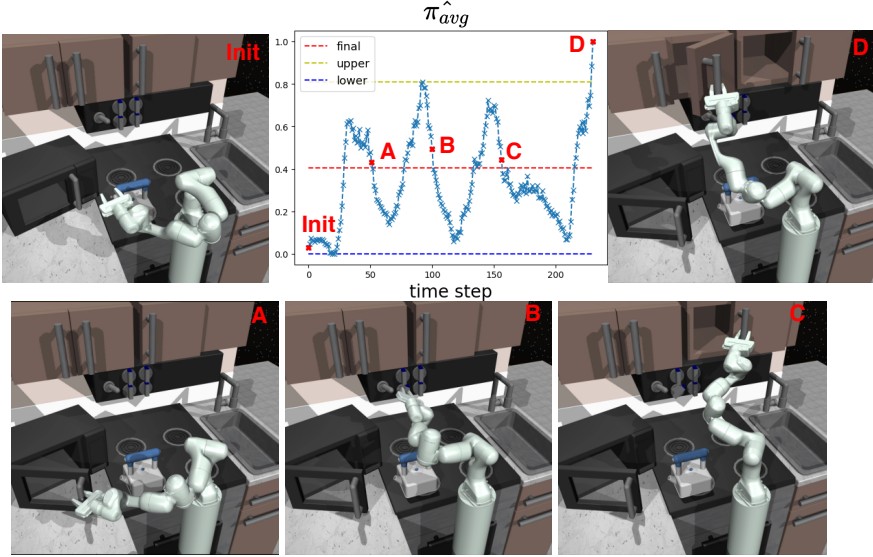

Figure 5: Visualization of the learnt expanding positions of Kitchen. The subtasks are Microwave, Bottom Knob, Hinge Cabinet and Slider. We show the upper bound, lower bound and final threshold produced by our detection algorithm.

## 4.4 ABLATION STUDY

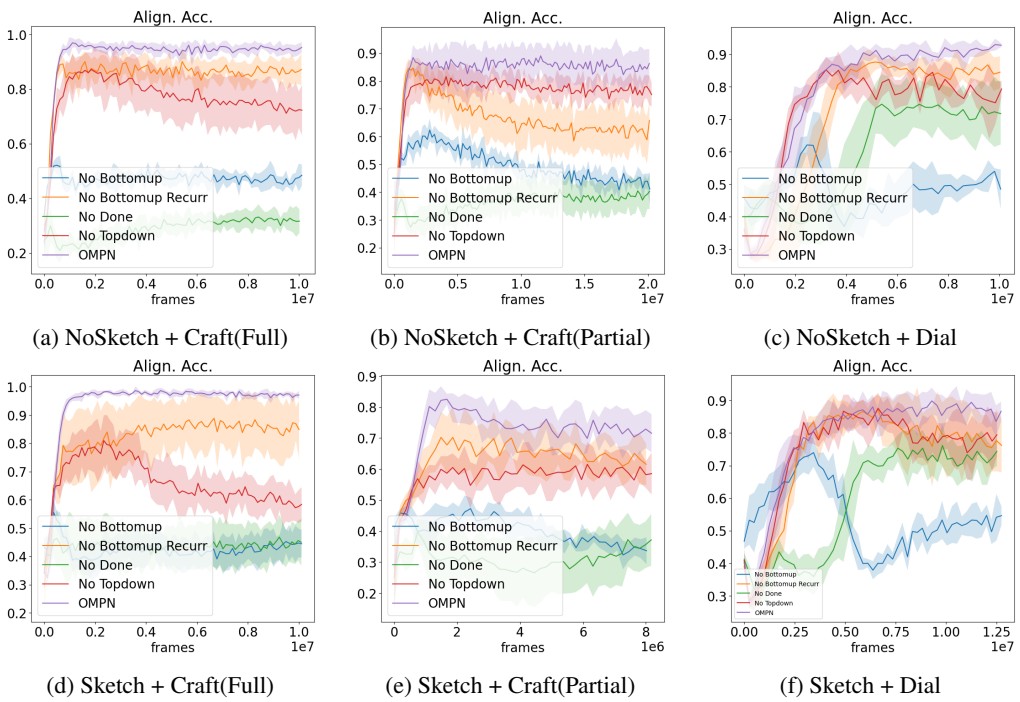

Figure 6: Ablation study for task alignment accuracy for all settings.

In this section, we aim to answer **Q3** and **Q4**. We perform the ablation study as well as some hyper-parameter analysis. The results are in Figure 6. We summarize our findings as below:

For *No Bottomup* and *No Topdown*, we remove completely either the bottom-up or the top-bottom process from the model. We find that both hurt the alignment accuracy and removing bottom-up process hurts more. This is expected since bottom-up recurrence updates the subtasks with the latest observations before predicting the termination scores, while the top-down process is more related to outputting actions.

For *No Bottomup Recurr*, we remove the recurrence in the bottom-up process by making $C_i^t = \mathcal{F}(\mathbf{0}, x^t, M_i^{t-1})$ so as to preserve the same number of parameters as OMPN. Although this hurts the alignment accuracy least, the existence of the performance drop confirms our intuition that the outputs of the lower-level subtasks are beneficial for the higher-level subtasks to predict termination scores, resulting in better task decomposition.

For *No Done*, we use the OMPN architecture but remove the $\pi_{end}$ from the loss. We find that the model is still able to learn the structure based on the inductive bias to some degree, but the alignment accuracy is much worse.

We also perform hyper-parameter analysis and see its effect on the task decomposition results in Appendix H. We find that our task decomposition results is robust to the memory depths $n$ and memory size $m$ in most cases.

### 4.5 BEHAVIOR CLONING

We show the behavior cloning results in Figure 7 and the full results are in Figure 10. For Craft, the sketch information is necessary to make the task solvable, so we don't see much difference when there is no sketch information. On the contrary, with full observation and sketch information (Figure 7a), this setting might be too easy to show the difference since a memory-less MLP can also achieve almost perfect success rate. As a result, only when the environment moves to a more challenging but still solvable setting with partial observation (Figure 7b), OMPN outperform LSTM on the success rate.

For Dial, we find that behaviour cloning alone is not able to solve the task and all of our models never generate the maximum return due to the exposure bias. This is consistent with the previous literature on applying behavior cloning to robotics tasks with continuous states (Laskey et al., 2017). More details can be found in Figure 13.

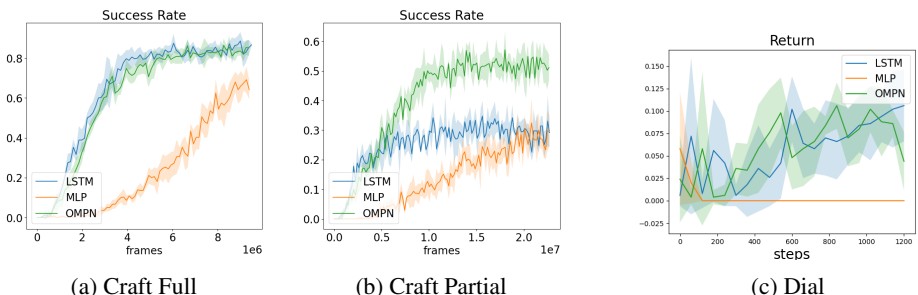

(a) Craft Full      (b) Craft Partial      (c) Dial

Figure 7: The behavior cloning results when sketch information is provided. For Craft, we define success as the completion of four subtasks. For Dial, the total maximum return is 4.

## 5 CONCLUSION

In this work, we investigate the problem of learning the subtask hierarchy from the demonstration trajectory. We propose a novel Ordered Memory Policy Network (OMPN) that can represent the subtask structure and leverage it to perform unsupervised task decomposition. Our experiments show that OMPN learns to recover the subtask boundary in both unsupervised and weakly supervised settings with behavior cloning. In the future, we plan to develop a novel control algorithm based on the inductive bias for faster adaptation to compositional combinations of the subtasks.

**Acknowledgement** This work is in part supported by the Center for Brain, Minds, and Machines (CBMM, funded by NSF STC award CCF-1231216), and IBM Research.

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

# A   OMPN ARCHITECTURE DETAILS

We use the gated recursive cell function from Shen et al. (2019) in the top-down and bottom up recurrence. We use a two-layer MLP to compute the score $f_i$ for the stick-breaking process. For the initial memory $M^0$, we send the environment information into the highest slot while keep the rest of the slots to be zeros. If unsupervised setting, then the every slot is initialized as zero. At the first time step, we also skip the bottom-up process and hard code $\pi^1$ such that the memory expands from the highest level. This is used to make sure that at first time step, we could propagate our memory with the expanded subtasks from root task. In our experiment, our cell functions does not share the parameters. We find that to be better than shared-parameter.

We set the number of slots to be 3 in both Craft and Dial, and each memory has dimension 128. We use Adam optimizer to train our model with $\beta_1 = 0.9, \beta_2 = 0.999$. The learning rate is 0.001 in Craft and 0.0005 in Dial. We set the length of BPTT to be 64 in both experiments. We clip the gradients with L2 norm 0.2. The observation has dimension 1076 in Craft and 39 in Dial. We use a linear layer to encode the observation. After reading the output $O^t$, we concatenate it with the observation $x^t$ and send them to a linear layer to produce the action.

In section 2, we describe that we augment the action space into $\mathcal{A} \cup \{done\}$ and we append the trajectory $\tau = \{s_t, a_t\}_{t=1}^T$ with one last step, which is $\tau \cup \{s_{t+1}, done\}$. This can be easily done if the data is generated by letting an expert agent interact with the environment as in Algorithm 1. If you do not have the luxury of environment interaction, then you can simply let $s_{T+1} = s_T, a_{T+1} = done$. We find that augmenting the trajectory in this way does not change the performance in our Dial experiment, since the task boundary is smoothed out across time steps for continuous action space, but it hurts the performance for Craft, since the final action of craft is usually $USE$, which can change the state a lot.

---

**Algorithm 1:** Data Collection with Gym API

---

env = gym.make(name)
done = False
obs = env.reset()
traj = []
**repeat**
    action = bot.act(obs)
    nextobs, reward, done = env.step(action)
    traj.append((obs, action, reward))
    obs = nextobs
**until** *done is True*;
traj.append((obs, done_action))

---

# B   THRESHOLDING ALGORITHM

We here provide the peseudo-code for our threshold algorithm. Algorithm 2, we detect boundary by recording the time steps which goes from above to below the given threshold. We return the final $K$ time steps where $K$ is given. We assume the last subtask ends when the episode ends.

We design Algorithm 3 to automatically select the threshold. Our intuition is that, the optimal threshold should pass through $K$ peaks, where $K$ is the provided by the users. As a result, we pick the highest peak and the lowest valley as the upper bound and lower bound for the threshold, and we return the final threshold as the middle of these two.

# C   TASK DECOMPOSITION METRIC

## C.1   F1 SCORES WITH TOLERANCE

For each trajectory, we are given a set of ground truth task boundary $gt$ of length $L$ which is the number of subtasks. The algorithm also produce $L$ task boundary predictions. This can be done in

---

**Algorithm 2:** Get boundary from a given threshold

---

**Data:** Standardized average expanding positions $\hat{\pi_{avg}}$ with length $L$, the number of subtasks $K$, and a threshold $T$
**Result:** A list of $K$ split points.
$preds = []$
$prev = False$
**for** $t$ *in range(L)* **do**
    $curr = [\hat{\pi_{avg}}[t] > T]$
    **if** $prev == curr$ **then**
        continue
    **else**
        **if** $prev$ *and not* $curr$ **then**
            $preds$.append($t - 1$)
        $prev = curr$
$preds$.append($L - 1$)
Return $preds[:: -1][: K][:: -1]$

---

---

**Algorithm 3:** Automatic threshold selection

---

**Data:** Standardized average expanding positions $\hat{\pi_{avg}}$ with length $L$
**Result:** A final threshold $T$
diff = $\hat{\pi_{avg}}$[1:] - $\hat{\pi_{avg}}$[:-1]
upid = [i for i in range(1, len(diff)) if diff[i] < 0 and diff[i-1] > 0]
lowid = [i for i in range(1, len(diff)) if diff[i] > 0 and diff[i - 1] < 0]
upval = $\hat{\pi_{avg}}$[upid]
lowval = $\hat{\pi_{avg}}$[lowid]
upper = upval.max()
lower = lowval.min()
final = (upper + lower) / 2
Return final

---

OMPN by setting the correct $K$ in $topK$ boundary detection. For compILE, we set the number of segments to be equal to $N$. Nevertheless, our definition of F1 can be extended to arbitaray number of predictions.

$$precision = \frac{\sum_{i,j} match(preds_i, gt_j, tol)}{\#predictions)}$$

$$precision = \frac{\sum_{i,j} match(gt_i, preds_j, tol)}{\#ground\ truth}$$

where the $match$ is defined as

$$match(x, y, tol) = [y - tol \leq x \leq y + tol]$$

where $[]$ is the Iverson bracket. The tolerance

## C.2 TASK ALIGNMENT ACCURACY

This metric is taken from Shiarlis et al. (2018). Suppose we have a sketch of 4 subtasks $b = [b1, b2, b3, b4]$ and we have the ground truth assignment $\xi_{true} = \{\xi_{true}^t\}_{t=1}^T$. Similar we have the predicted alignment $\xi_{pred}$. The alignment accuracy is simply

$$\sum_t [\xi_{pred}^t == \xi_{true}^t]$$

For OMPN and compILE, we obtain the task boundary first and construct the alignment as a result. For TACO, we follow the original paper to obtain the alignment.

## D BASELINE

### D.1 COMPILE DETAILS

| latent | [concrete, gaussian] |
|---|---|
| prior | [0.3, 0.5,0.7] |
| kl_b | [0.05, 0.1, 0.2] |
| kl_z | [0.05, 0.1, 0.2] |

Table 2: compILE hyperparameter search.

Our implementation of compILE is taken from the author github[4]. However, their released code only work for a toy digit sequence example. As a result we modify the encoder and decoder respectively for our environments. During our experiment, we perform the following hyper-parameter sweep on the baseline in Table 2. Although the authors use latent to be concrete during their paper, we find that gaussian perform better in our case. We find that Gaussian with $prior = 0.5$ performs the best in Craft. For Dial, these configurations perform equally bad.

We show the task alignments of compILE for Craft in Figure 8. It seems that compILE learn the task boundary one-off. However, since the subtask ground truth can be ad hoc, this brings the question how should we decide whether our model is learning structure that makes sense or not? Further investigation in building a better benchmark/metric is required.

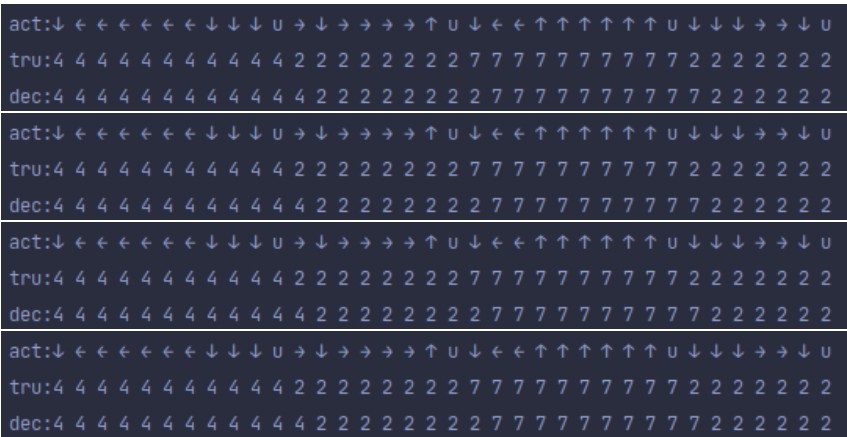

Figure 8: Task alignment results of compILE on Craft.

### D.2 TACO DETAILS

| dropout | [0.2, 0.4, 0.6, 0.8] |
|---|---|
| decay | [0.2, 0.4, 0.6, 0.8] |

Table 3: TACO hyperparameter search.

We use the implementation from author github[5] and modifiy it into pytorch. Although the author also conduct experiment on Craft and Dial, they did not release the demonstration dataset they use. As a result, we cannot directly use their numbers from the paper. We also apply dropout on the prediction of $STOP$ and apply a linear decaying schedule during training. The hyperparameter search is in table 3. We find the best hyperparameter to be $0.4, 0.4$ for Craft. For Dial, the result is not sensitive to the hyperparameters.

---

[4] https://github.com/tkipf/compile
[5] https://github.com/KyriacosShiarli/taco

# E    DEMONSTRATION GENERATION

We use a rule-based agent to generate the demonstration for both Craft and Dial. For Craft, we train on 500 episodes each on $MakeAxe$, $MakeShears$ and $MakeBed$. Each of these task is further composed of four subtasks. For Dial, we generate 1400 trajectories for imitation learning with each sketch being 4 digits.

For Craft with full observations, we design a shortest path solver to go to the target location of each subtask. For Craft with partial observation, we maintain an internal memory about the currently seen map. If the target object for the current subtask is not seen on the internal memory, we perform a left-to-right, down-to-top exploration until the target object appears inside the memory. Once the target object is seen, it defaults to the behavior in full observations. For Dial, we use the hand-designed controller in Shiarlis et al. (2018) to generate the demonstration.

# F    CRAFT

We train our models on $MakeBed$, $MakeAxe$, and $MakeShears$. The detail of their task decomposition is in Table 4. We show the behavior cloning results for all settings in Figure 10. We display more visuzliation of task decomposition results from OMPN in Figure 9.

| makebed | get wood, make at toolshed, get grass, make at workbench |
| makeaxe | get wood, make at workbench, get iron, make at toolshed |
| makeshears | get wood, make at workbench, get iron, make at workbench |

Table 4: Details of training tasks decomposition.

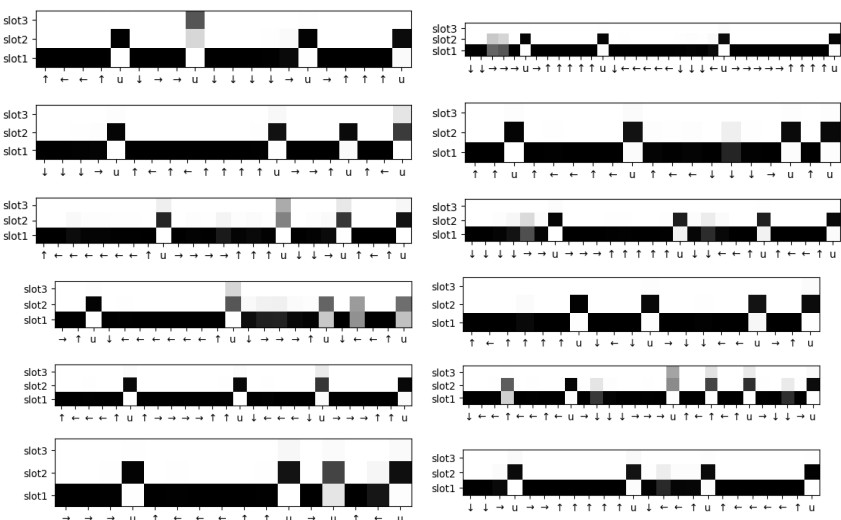

Figure 9: More results on $\pi$ in Craft. The model is able to robustly switch to a higher expanding position at the end of subtasks. The model will also sometimes discover multi-level hierarchy.

We show the results of task decomposition when the given $K$ is different in table 5 and table 6. We find that when you increase the $K$, the recall increases while the precision decreases.

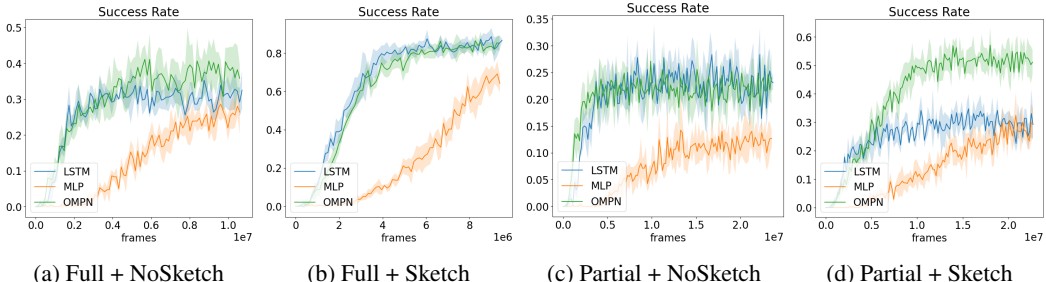

(a) Full + NoSketch    (b) Full + Sketch    (c) Partial + NoSketch    (d) Partial + Sketch

Figure 10: Behavior Cloning for Craft.

| | Full NoSketch | | | Full Sketch | | |
|---|---|---|---|---|---|---|
| | F1(tol=1) | Pre(tol=1) | Rec(tol=1) | F1(tol=1) | Pre(tol=1) | Rec(tol=1) |
| K=2 | 62(2.4) | 93(3.7) | 47(1.8) | 62(2.4) | 93(3.4) | 46(1.8) |
| K=3 | 81(2) | 95(2.4) | 71(1.8) | 81(2.2) | 95(2.3) | 71(2.0) |
| K=4 | 95(1.2) | 95(1.8) | 95(1.8) | 98(0.9) | 98(1.6) | 97(2.1) |
| K=5 | 92(1.6) | 87(2.8) | 99(0.5) | 93(6.4) | 87(1.1) | 99(0.5) |
| K=6 | 88(2.9) | 78(3.9) | 100(0) | 88(0.6) | 79(0.9) | 99(0.2) |

Table 5: Parsing results for full observations with different $K$

| | Parital NoSketch | | | Partial Sketch | | |
|---|---|---|---|---|---|---|
| | F1(tol=1) | Pre(tol=1) | Rec(tol=1) | F1(tol=1) | Pre(tol=1) | Rec(tol=1) |
| K=2 | 64(1.8) | 96(2.7) | 48(1.4) | 57(3.7) | 88(5.6) | 43(2.8) |
| K=3 | 83(1.9) | 96(1.9) | 72(1.9) | 74(3.9) | 88(4.6) | 64(3.5) |
| K=4 | 89(4.6) | 97(1.6) | 82(2.7) | 83(8.1) | 88(4.4) | 80(4.6) |
| K=5 | 93(0.8) | 90(1.9) | 96(2.7) | 89(3.2) | 85(3.7) | 94(3.8) |
| K=6 | 90(1.9) | 85(1.7) | 98(2.3) | 87(2.1) | 813.4) | 97(2.3) |

Table 6: Parsing results for partial observations with different $K$

# G    DIAL

We show in Table 7 the task alignment result for different thresholds. We can see that optimal fixed threshold is around 0.4 or 0.5 for Dial, and our threshold selection algorithm could produce competitive results. We demonstrate more expanding positions in Figure 11. We also show the failure cases, where our selected threshold fails to recover the skill boundary. Nevertheless, we can see that the peak pattern exists. We show the task alignment curves for different thresholds as well as the auto-selected threshold in Figure 12.

|  | Align. Acc. at different threshold | | | | | | |
|---|---|---|---|---|---|---|---|
|  | 0.2 | 0.3 | 0.4 | 0.5 | 0.6 | 0.7 | Auto |
| OMPN + noenv | 57(12.8) | 76(9.5) | 87(5.7) | 89(1.4) | 81(5.7) | 70(9.7) | 87(4) |
| OMPN + sketch | 71(11.6) | 81(10.6) | 85(7.4) | 84(6.6) | 76(7.7) | 60(6.4) | 84(5.7) |

Table 7: Task alignment accuracy for different threshold in Dial. The result for automatic threshold selection is in the last column.

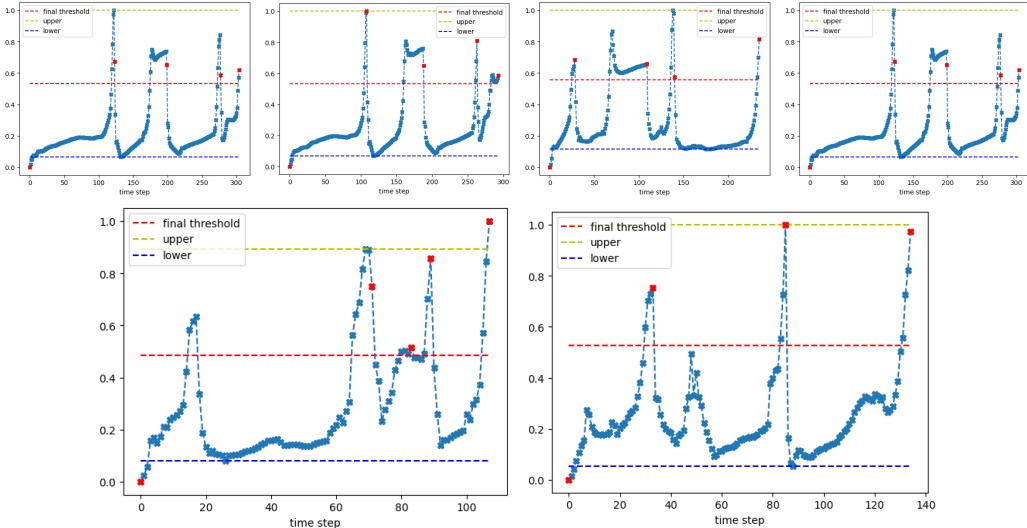

Figure 11: More task decomposition visualization in Dial. Our algorithm recovers the skill boundary for the first four trajectories but fails in the last two. Nevertheless, one can see that our model still display the peak patterns for each subtask, and a more advanced thresholding method could be designed to recover the skill boundaries.

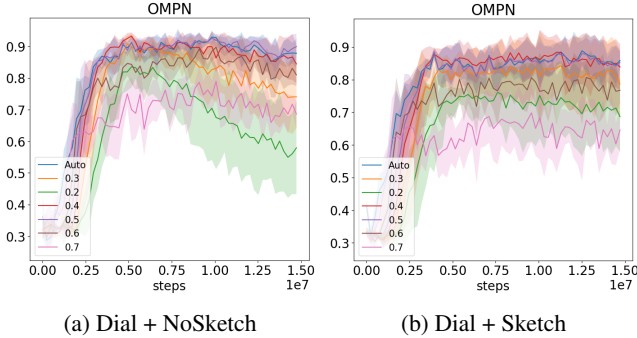

(a) Dial + NoSketch          (b) Dial + Sketch

Figure 12: The learning curves of task alignment accuracy for different thresholds as well as the automatically selected one.

The behavior cloning results are in Figure 13.

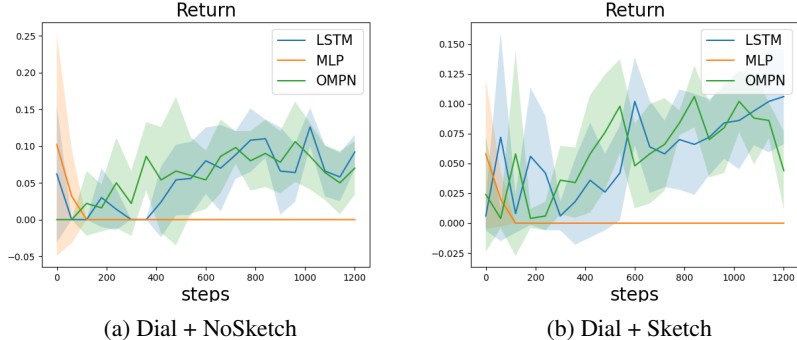

(a) Dial + NoSketch            (b) Dial + Sketch

Figure 13: The behavior cloning results in Dial domain.

We show the task decomposition results when the $K$ is mis-specified in Table 8.

|  | NoSketch | | | Sketch | | |
|---|---|---|---|---|---|---|
|  | f1(tol=1) | rec(tol=1) | pre(tol=1) | f1(tol=1) | rec(tol=1) | pre(tol=1) |
| $K=2$ | 59(2.9) | 45(2.1) | 90(4.4) | 58(1.4) | 44(1.1) | 88(2.1) |
| $K=3$ | 73(5.5)) | 63(5.1) | 85(6.2) | 72(3.8) | 63(3.4) | 84(4.5) |
| $K=4$ | 84(7.1) | 81(7.1) | 84(6.9) | 81(4.8) | 81(4.6) | 82(5.1) |
| $K=5$ | 86(5.1) | 91(3.6) | 83(6.3) | 83(5.2) | 86(5.4) | 82(5.3) |
| $K=6$ | 87(4.2) | 93(1.5) | 82(6.2) | 84(5.2) | 88(5.5) | 81(5.2) |
| $K=7$ | 87(4.1) | 93(1.4) | 82(6.4) | 84(5.2) | 89(6) | 81(5.2) |
| $K=8$ | 86(4.1) | 93(1.6) | 82(6.6) | 84(5.2) | 90(6..1) | 81(5.1) |

Table 8: F1 score, recall and precision with tolerance 1 computed at different $k$.

# H  HYPERPARAMETER ANALYSIS

| | Craft Full | | Craft Partial | | Dial | |
|---|---|---|---|---|---|---|
| | NoSketch | Sketch | NoSketch | Sketch | NoSketch | Sketch |
| $n = 3, m = 64$ | 93(1.7) | 97(1.2) | 84(6) | 72(6.2) | 87(4) | 82(7.4) |
| $n = 2, m = 64$ | 96(1.4) | 96(1.4) | 87(3.3) | 77(1.2) | 88(3.2) | 82(5.7) |
| $n = 4, m = 64$ | 96(0.6) | 96(2.5)) | 88(3.1) | 78(10.7) | 86(9.8) | 84(10) |
| $n = 3, m = 32$ | 92(4.2) | 91(10.3) | 88(6) | 74(5.5) | 88(4.9) | 83(3.7) |
| $n = 3, m = 128$ | 96(1.5) | 97(1.2) | 86(3) | 75(5.7) | 87(2.4) | 83(6.2) |

Table 9: Task alignments accuracy for different memory dimension $m$ and depths $n$. The default setting is on the first row. The next two rows change the depths, while the last two rows change the memory dimension. The number in the parenthesis is the standard deviation.

We discuss the effect of hyper-parameters on the task decomposition results. We summarize the results in Table 9. We show the detailed alignment accuracy curves for Craft in Figure 14 and for Dial in Figure 15.

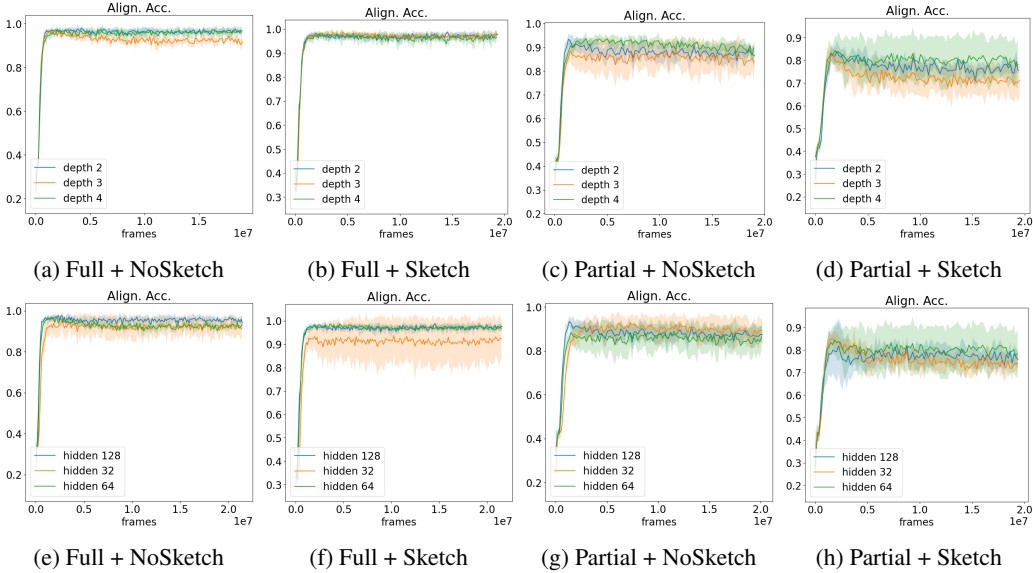

Figure 14: Task alignment accuracy with varying hyperparameters in Craft. We change the depths in the first row and memory size in the second row.

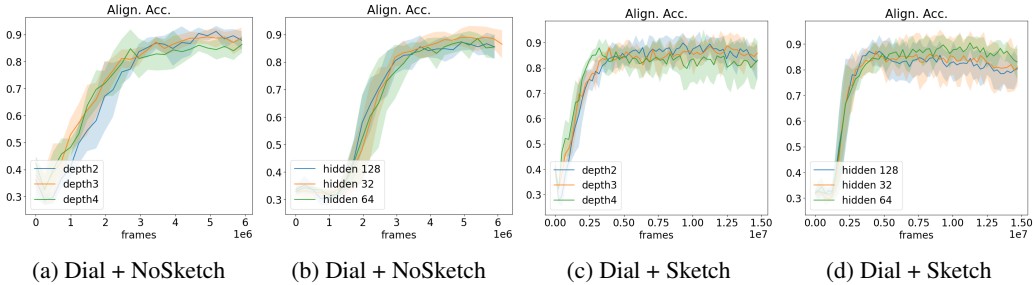

Figure 15: Task alignment accuracy in Dial. In (a) and (c) we change the depths while in (b) and (d) we change the memory size.

# I QUALITATIVE RESULTS ON KITCHEN

We display the qualitative results for Kitchen. We use the demonstration released by Gupta et al. (2019). Since they do not provide ground truth task boundaries, we provide the qualitative results. We use a hidden size of 64 and the memory depths of 2. We set the learning rate to be 0.0001 with Adam and the BPTT length to be 150. The input dimension is 60 and action dim is 9. We train our model in the unsupervised (NoSketch) setting. We show the visualization of our expanding position, in a similar fashion of the Dial domain, in Figure 16 and Figure 17. We show the visualizations for other trajectories in Figure 18

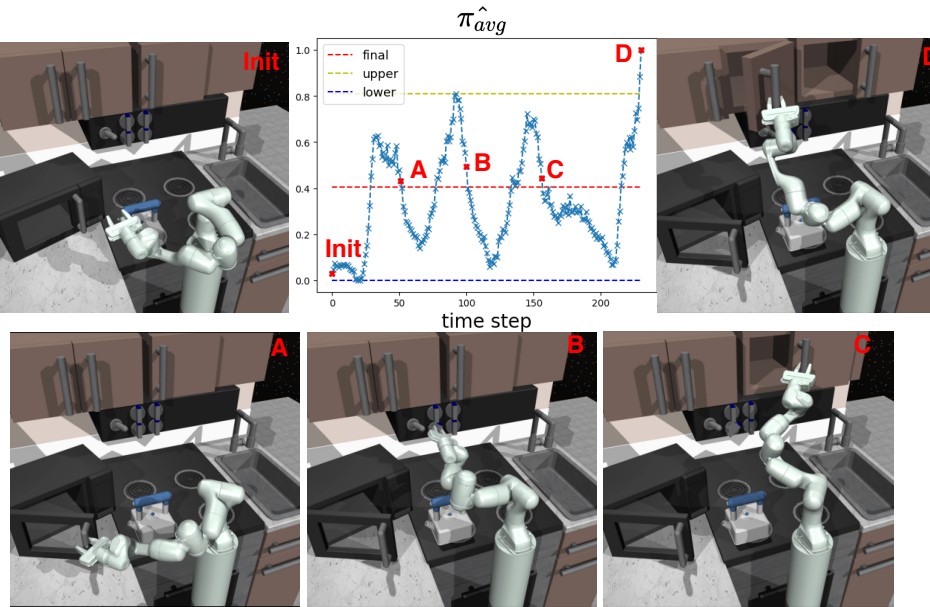

Figure 16: Visualization of the learnt expanding positions of Kitchen. The subtasks are Microwave, Bottom Knob, Hinge Cabinet and Slider.

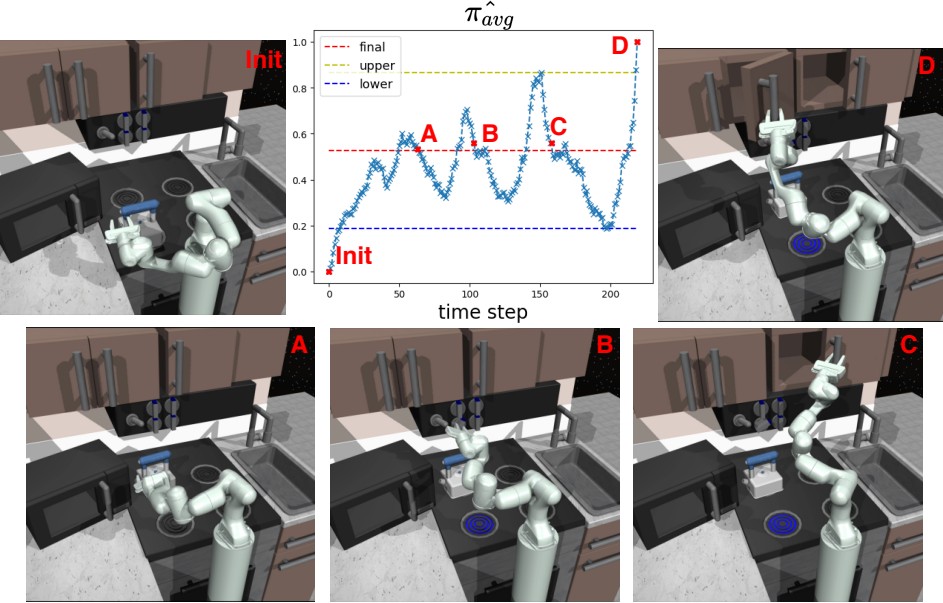

Figure 17: Visualization of the learnt expanding positions of Kitchen. The subtasks are Kettle, Bottom Knob, Hinge Cabinet and Slider.

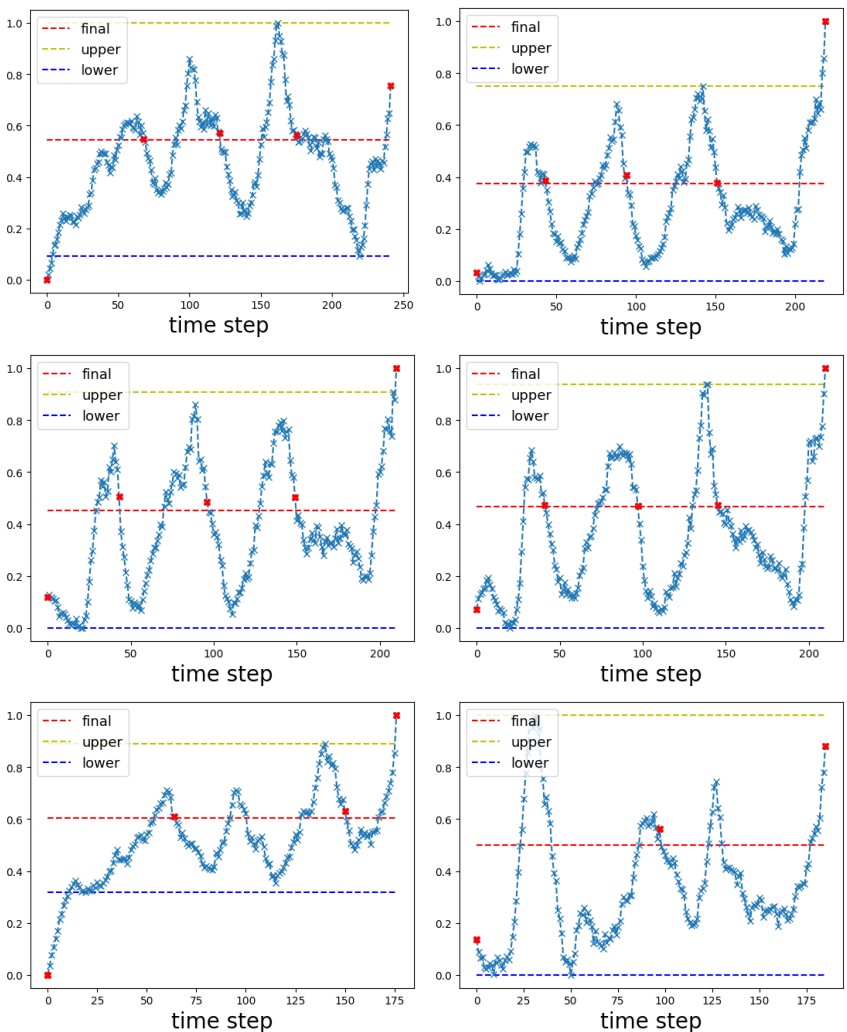

Figure 18: More task decomposition visualization in Kitchen. Our algorithm seems to recover the skill boundary for the first four trajectories but fails in the last two. Nevertheless, one can see that our model still display the peak patterns for each subtask, and a more advanced post-processing thresholding method might be able to recover the task boundary.

