# OpenReview forum: "Learning Task Decomposition with Ordered Memory Policy Network"
_ICLR.cc/2021/Conference — ICLR 2021 Poster_

### Official Review · AnonReviewer4 · 2020-10-21
**Interesting model, but the paper requires some minor changes**

**Rating:** 6
**Confidence:** 3

**Review:**

*Summarize what the paper claims to contribute. Be positive and generous.*

The authors introduce a new architecture, the Ordered Memory Policy Network (OMPN) that has an explicit inductive bias for modelling a hierarchy of sub-tasks. They claim that OMPN discovers task decomposition on demonstration from imitation learning, both in an unsupervised and weakly supervised setting, and its performance compares favourably with that of strong baselines on the Craft and Dial tasks. The paper also includes a small ablation study that illustrates the effect on performance of removing either bottom-up or top-down recurrence from the network. There is some analysis of performance, showing that the model can indeed identify the sub-task boundaries in both the Craft and Dial tasks.

*List strong and weak points of the paper. Be as comprehensive as possible.*

**Strong points**
- I find the architecture that the authors propose interesting and novel. There is a comprehensive comparison with other approaches in the literature and a fair quantitative comparison with relevant baselines on two benchmarks.
- The authors provide a simple ablation study to show how important are the top-down or bottom-up recurrent connections in their model. They also provide some visualisation to show how the model segments the sequence into sub-tasks.

**Weak points**

- I found section 2.1 and 2.2 a bit dense and slow to read. I appreciate it might be difficult to communicate a novel and complex model effectively to every reader, but though I could understand the various components, I did not feel I had gained much insight into why this model makes sense. Perhaps the authors could try to communicate their motivation for the design choices in a revised version of the paper - but I leave this up to the authors.
- Though the paper provides a quantitative comparison with baselines on two datasets, it would be great to know what the authors think are the limitations of their proposed model. Will it scale? In what cases/condition will it not work?
- For experiments in Craft, it is not mentioned how the experience is collected. Please include a clear description or note mentioning this.

*Clearly state your recommendation (accept or reject) with one or two key reasons for this choice. Provide supporting arguments for your recommendation.*

I recommend that the paper is conditionally accepted. The paper presents a novel method that outperforms other baselines on a difficult problem and does provide some analysis as to how the model works and some ablation results. Overall, the paper is well written, but there are a few points that I believe need to be addressed, if the paper is to be accepted. Please see minor comments below for small typos and necessary additions. Another condition is that the authors specify how the trajectories in Craft were collected (see last point of weak points above).

*Ask questions you would like answered by the authors to help you clarify your understanding of the paper and provide the additional evidence you need to be confident in your assessment. *

Please respond to my points above. In addition, I would be interested to understand a bit more the insight of how the model works and what motivates the choices you made. Also, I believe that Figure 3 requires some more explanation, as I am not confident I understand everything that’s meant to communicate.

*Provide additional feedback with the aim to improve the paper. Make it clear that these points are here to help, and not necessarily part of your decision assessment.*

- Figure 4: Please indicate which task id/level correspond to each subplot, or means to reproduce it. Either to the plots or the figure caption.
- Page 3: “The score fti can be interpreted **as** the probability that subtask i is completed at time t.”
- Page 4: “We find this method is suitable for continuous control ~~settubgs~~ settings, where the subtask boundaries are more ambiguous and smoothed out ~~accross~~ across time steps.”
- Page 5: “However ONLSTM does not ~~provides~~ provide mechanism to achieve the top-down and bottom-up recurrence.”

---

> ### Author Response · Authors · 2020-11-16
> **Response to Reviewer 4**
>
> Thank you for your insight, and we are glad to hear that you find our idea to be interesting and our comparisons to be comprehensive, fair and quantitative. Here is our response to your concerns.
>
> > I found section 2.1 and 2.2 a bit dense and slow to read. I appreciate it might be difficult to communicate a novel and complex model effectively to every reader, but though I could understand the various components, I did not feel I had gained much insight into why this model makes sense.
>
> We are sorry for the confusion, and we thank you for your understanding. We will rewrite it to make it more readable. Our main goal is to design an architecture that can operate with a subtask hierarchy like our toy example in Fig1. We approach this problem in two folds:
>
> - In section 2.1, we focus on designing an architecture to support a subtask hierarchy while also maintaining the end-to-end property. We focus on replacing each concept in Fig1 with a differentiable operation along with proper inductive bias (e.g., stick breaking process).
>
> - In section 2.2, we further try to regularize our model during the optimization process. We observed that when excluding pi_end, the model degenerated into only using the low-level slot and not learning any structure, while adding pi_end avoided such degeneration, leading to better F1 score and task alignment accuracy. We speculate that adding explicit pi_end helps regularize our model to not always output a low expanding position, because predicting the pi_end requires a very high expanding position by our design. This regularization is also general for each environment.
>
> > Though the paper provides a quantitative comparison with baselines on two datasets, it would be great to know what the authors think are the limitations of their proposed model. Will it scale? In what cases/condition will it not work?
>
> - Our method is designed to be scalable, since we make sure that our model is an end-to-end trainable architecture. Our model is designed to be an off-the-shelf memory module, which can be used to replace the LSTM component in the more complex environments [1].
>
> -  One potential limitation might come with the fact that our model is designed to be general. Then according to the no free lunch theorem, in some specific narrow domains, there might exist a tailored algorithm that works better.
> In the future, we also plan to propose a HRL method, which can utilize the skill knowledge gained in the imitation learning phase.
>
> > For experiments in Craft, it is not mentioned how the experience is collected. Please include a clear description or note mentioning this.
>
> Our demonstrations, for both Craft and Dial, are generated by a hand-designed rule-based model. We did not create the demo from an RL agent, because we think that might give a hidden advantage to the IL agent with the same architecture as the RL agent, making the comparisons unfair. As a result, we specifically try to avoid it by using a rule-based model.
>
> For Craft with full observation, the demonstration is generated by planning a shortest path to finish each subtask. For Craft with partial observation, we keep an internal memory about the portion of map that we’ve seen. For each subtask, if the target is already seen, it defaults to the behavior of full observation. If not seen, then the agent performs exploration, until the target object is found. For Dial, we use the hand craft controller implemented at https://github.com/KyriacosShiarli/taco. We will clarify these details in our revision.
>
> We will fix the typos and polish the figures and tables accordingly. Please let me know if you have more questions. We are more than happy to solve your concerns.
>
> [1] Berner, Christopher, et al. "Dota 2 with large scale deep reinforcement learning." arXiv preprint arXiv:1912.06680 (2019).

---

> > ### Comment · AnonReviewer4 · 2020-11-20
> > **Satisfied with authors' response**
> >
> > Thank you for your detailed response and for taking the time to address every single point. Out of interest I would like to ask, which would be a challenging domain, or a real world application, that you want to try your proposed model? Why choose that domain, and what advantages do you expect your method to have compared to other approaches currently used on that domain?

---

> > > ### Author Response · Authors · 2020-11-23
> > > **About Challenging domains**
> > >
> > > Thank you for your interest. We believe that our model is a natural fit for the robotics domain (e.g., Kitchen), where reusing the previously learnt subtasks is important for faster adaptation in new tasks. To be specific,
> > >
> > > 1. During hierarchical imitation learning stage, we use the behaviour cloning in this paper to learn the subtask hierarchy.
> > >
> > > 2. During RL stage on potentially new tasks, we could imagine using the learnt expanding position in step1 as an auxilliary reward to help an agent to reach the subgoal states during the exploration. This should help the agent to adapt faster in a sparse reward setting compared with other methods.
> > >
> > > We just updated the paper, and we are looking forward to your further feedbacks.

---

> ### Comment · Area_Chair1 · 2020-11-18
> **Author response**
>
> Dear AnonReviewer4,
>
> We are now entering the second discussion stage. Could you please check whether the authors have addressed your concerns and questions and potentially ask any further clarification questions?
>
> Thank you,
> Your Area Chair

---

### Official Review · AnonReviewer3 · 2020-10-28
**Review: Learning task decomposition with order-memory policy network**

**Rating:** 4
**Confidence:** 2

**Review:**

#######################################################################

Summary:

In this paper the authors propose a new method for task decomposition. First a new neural architecture is proposed which includes a set of 'memories' who's operation is inspired by the HAMs architecture in which individual memories correspond to subtasks which can be internally updated, call the next-level subtask in the stack, or return control to the operating subtask. After training, subtask boundaries can then be extracted by inspecting the values of these memories.

#######################################################################

Reasons for score:

While I think the proposed method is interesting concept there seems to me to be insufficient evidence to suggest that it is general. The authors seem to explicitly control the depth of the memory stack and the expected number of subtasks in subsequent analysis but do not present alternate results. I think the current work would be notably stronger given broader experimental support.

#######################################################################

Pros:

1. I found the introduction to be clearly written (if lightly referenced) and appreciated the simple example

2. The architectural construction itself would seem to be reasonably simple, and therefore potentially more broadly applicable in the future.

#######################################################################

Cons:

1. There were a number of unjustified (or weakly justified) design choices made throughout the paper. While I understand that this is to some extent inevitable when implementing new methods, I think the paper could be notably bolstered by discussing these further. For example I am not sure how the number of memories 'n', or the dimension of those memories 'm' ought to be chosen in general, or what the effect of different choices might be. Similarly, how was the threshold value selected? Simply by visual inspection? It would also be valuable to expound on the rational for making the bottom-up relation recurrent.

2. The experimental section could be strengthened

(1) nit: Fig 3 would be much easier to read if you used images rather than letters in your plots

(2) nit: In general all axes should be labeled, experiments separated by title, methods by colour etc

(3) Experiments are insufficient to justify claims (for example, in the craft world partially observable setting, the observed performance improvement over LSTM seem to be significant - it'd have been great to see that further explored... for the top-k choice, why not show the results for other values of k or suggest methodology for determining this value automatically)

#######################################################################

Questions during rebuttal period:

Q1: if a practitioner does not know the correct value of 'n' and 'k' ahead of time, how would they use this approach to discover the task decomposition? Or rather, how should these values be chosen?

Q2: could you say a bit more about the degenerate behaviour observed when the explicit pi_end was excluded?

Q3: the alignment accuracy for different thresholds in Fig9 are not monotonic - do you have any intuition for why this is?

#######################################################################

Some typos:

(1) nit: bolding key / leading values in table has probably become standard now

(2) Fig 6 axes values

---

> ### Author Response · Authors · 2020-11-16
> **Response to Reviewer 3 (Part 1)**
>
> Thank you for your feedback, and we are glad to hear that you find our idea to be interesting and our toy example to be intuitive. Our central message is that OMPN is a general off-the-shelf model for task decompositions.
>
> > The authors seem to explicitly control the depth of the memory stack and the expected number of subtasks in subsequent analysis but do not present alternate results. I think the current work would be notably stronger given broader experimental support.
>
> >  How should a practioner choose 'n', 'k' and 'm'?
>
> We will add the hyperparameter analysis in our revision to provide further discussion. Right now describe the guideline for practioners to select these values:
>
> - The practioner does not need to know $k$ to train our model, which is our biggest advantage over the existing literature. However, the practioner needs to specify $k$ during the detection phase only, but this is the setting in our baseline [1] as well so we follow it for a fair comparison. Furthermore, the practoner can have a good guess of $k$ by visual inspection after training. The inferred value of $k$ can be used in the downstream detection algorithm.
>
> - For the depth $n$, in practice, since most of the RL benchmarks focus on just two or three levels, we find that setting the number from 2 or 4 seems to be robust.
>
> - For the memory dimension $m$, a practioner could set the standard number as those in LSTM, e.g., 64, 128.
>
> > Similarly, how was the threshold value selected? Simply by visual inspection?
>
> We thank the reviewer for pointing this out. Our current version focuses on demonstrating the effectiveness of the thresholding. We now have an algorithm to automatically select the threshold. The idea is that a successful threshold should cut through all the peaks as is shown in the visualization. As a result, we pick the highest peak value as the upper bound, pick the lowest valley value as the lower bound, and compute the final threshold by taking an average of the two.
>
> > What is the rationale of bottom-up recurrence?
>
> Bottom up recurrence can return the final state of a lower-level subtask to a higher-level task. It's like returning the output value of a function (e.g. lower-level subtask) to its caller (high-level subtask). In addition to the observation, seeing the results returned by the lower-level subtask could be useful for the high-level subtask to understand better when to terminate, leading to better task decomposition. We would perform the ablation study for this in the revision.
>
> > could you say a bit more about the degenerate behaviour observed when the explicit pi_end was excluded?
>
> We observed that when excluding pi_end, the model degenerated into only using the low-level slot and not learning any structure, while adding pi_end avoided such degeneration, leading to better F1 score and task alignment accuracy. We speculate that adding explicit pi_end helps regularize our model to not always output a low expanding position, because predicting the pi_end requires a very high expanding position by our design.
>
> [1] Kipf, Thomas, et al. "CompILE: Compositional imitation learning and execution." International Conference on Machine Learning. PMLR, 2019.

---

> ### Author Response · Authors · 2020-11-16
> **Response to Reviewer 3 (Part 2)**
>
>
> > the alignment accuracy for different thresholds in Fig9 are not monotonic - do you have any intuition for why this is?
>
> The reason can be found in the visualization of expanding positions in Fig8 and Fig6. A good threshold should be the one that can cut through all the "peaks" in order to correctly identify the boundary. However, we find that not all peaks have the same height. As a result, when the threshold is too high, you will miss some peaks and the task alignment is worse. Similarly, when the threshold is too low, you would also miss some peaks. Hence, you have our observation in Fig9 that the optimal threshold is the intermediate one, which is between 0.4~0.6. We now have an algorithm to select the threshold automatically, and we will add it in the revision
>
>
> > In the craft world partially observable setting, the observed performance improvement over LSTM seem to be significant - it'd have been great to see that further explored…
>
> There are several reasons to answer why the gain is significant only in weakly supervised case with partial observation.
> - For the unsupervised (Fig4a & 4c) setting in general, the agent is not provided any sketch information during testing and thus the environment is not solvable. Hence it would be reasonable to explain that all the policy models perform bad on it.
>
> - In the weakly supervised setting with sketch information, the full observation (Figure 4b) is too easy. Even a memory-less MLP can approach an 80% success rate. Thus it also fails to show the advantage of our model as well.
>
> - As a result, weakly supervised + partial observation (Figure 4d) is a harder but still solvable environment. The partial observation requires the agent to store the information in the long-term memory. Our hypothesis is that the design of hierarchical subtask design helps preserve long term information.
>
> Additional: We will fix the tables and figures accordingly in our revision.

---

> ### Comment · Area_Chair1 · 2020-11-18
> **Author response**
>
> Dear AnonReviewer3,
>
> We are now entering the second discussion stage. Could you please check whether the authors have addressed your concerns and questions and potentially ask any further clarification questions?
>
> Thank you,
> Your Area Chair

---

> ### Author Response · Authors · 2020-11-21
> **Looking forward to your feedback**
>
> Dear Reviewer3,
>
> Thanks again for your constructive review, which has helped us improved the quality and clarity of the paper. In addition to our response above, in the revision, we have included comparisons with additional baselines and increased the complexity of the scene.
>
> As the discussion period is about to end, please don’t hesitate to let us know if there are any additional clarifications that we can offer, as we would love to convince you of the merits of the paper. We appreciate your suggestions. Thanks!

---

> ### Author Response · Authors · 2020-11-23
> **Revision Updated**
>
> Dear Reviewer,
>
> We have updated our paper, and we are looking forward to your further feedbacks.

---

### Official Review · AnonReviewer1 · 2020-10-28
**Interesting memory architecture for task decomposition but requires more thorough experiments**

**Rating:** 6
**Confidence:** 4

**Review:**

### Summary
This paper proposes a hierarchical memory structure, where each layer of memory stores information about the corresponding level of sub-task and uses this structure for better behavioral cloning. The memory module mimics the call-return mechanism, in which the higher-level memory (sub-task) changes to the next one when the lower-level memory (sub-task) is completed. With this inductive bias, the memory module and policy can be trained end-to-end from a sequence of state-action pairs, and learn effective task decomposition in multiple levels. By looking at changes in each level of the memory module, which means changes in sub-tasks, we can find the task decomposition. The experimental results on the Craft World (grid world) and Jaco arm dialing environments demonstrate that the proposed method can learn better task decomposition in both unsupervised and weakly supervised settings.

### Strengths
- The idea of using the ordered memory module to learn the subtask structure in multiple levels is novel and intuitive.
- The subtask decomposition results show superior task alignment of the proposed method over baseline methods in both unsupervised and weakly supervised learning settings.
- The paper is well-written and easy to follow.

### Weaknesses
- In Section 4.1, the paper claims that the proposed method outperforms the LSTM baseline due to its superior ability to store longer-term information. But, OMPN outperforms the LSTM baseline only under the weakly supervised setting. Is there any good explanation of why the proposed method does not show improved performance in the unsupervised setting?
- The learning curves and final performance in the Dial task are missing in the paper. Does it achieve higher performance compared to baselines (MLP, LSTM policies)?
- It is not clear whether the hierarchy in memory is important or not. For example, in Figure 3, the horizontal expansion does not happen in slot 3, which can mean 2 level-hierarchy could be enough for solving the task. Ablation study on a different number of levels in the memory module can help to understand the importance of the hierarchical memory.
- In Figure 6, when the robot arm presses the number 4, the expected expansion position is high for multiple steps, which means the high-level task is likely to be changed over multiple steps. But this result seems not desirable. Is there any explanation why this still makes sense?

### Questions and additional feedback
- It would be easier to understand Figure 3 if each map is rendered with icons, not with words.
- Adding sub-captions for four graphs in Figure 4 will make it easy to read.
- In Figure 5, what is the difference between (a,b) and (c,d)? It is unclear from the text and figure.

### Overall assessment
The proposed method is novel and intuitive and shows improved task decomposition on the Craft World and Dial environments. However, the paper needs a more thorough analysis of the results, and an additional ablation study can make the claims stronger. In summary, the reviewer thinks this paper can be accepted with a few more experiments and analysis.

---

> ### Author Response · Authors · 2020-11-16
> **Response to reviewer1**
>
> Thank you for your feedback. We are glad to hear that you find our idea to be novel and interesting and our paper to be well-written. Here is the response to your question:
>
> > However, the paper needs a more thorough analysis of the results, and an additional ablation study can make the claims stronger. In summary, the reviewer thinks this paper can be accepted with a few more experiments and analysis.
>
> We thank reviewer for pointing this out. We will update more ablations and hyper-parameter analysis to make our claim stronger in the revision. We also plan to add qualititive results on another more challenging environment Kitchen.
>
> > OMPN outperforms the LSTM baseline only under the weakly supervised setting. Is there any good explanation of why the proposed method does not show improved performance in the unsupervised setting?
> Yes. There are several reasons to explain why it happens:
> - For the unsupervised (Fig4a & 4c) setting, the agent is not provided any sketch information during testing and thus the environment is not solvable. Hence it would be reasonable to explain that all the policy models perform bad on it.
> - In the weakly supervised setting with sketch information, the full observation (Figure 4b) is too easy. Even a memory-less MLP can approach an 80% success rate. Thus it also fails to show the advantage of our model.
> - As a result, weakly supervised + partial observation (Figure 4d) is a harder but still solvable environment. The partial observation requires the agent to store the information in the long-term memory. Our model shows most gain here since the main motivation of hierarchical subtask design is to preserve long term information.
>
> > The learning curves and final performance in the Dial task are missing in the paper. Does it achieve higher performance compared to baselines (MLP, LSTM policies)?
>
> We thank the reviewer for pointing this out. We will add this in the revision.
>
> > It is not clear whether the hierarchy in memory is important or not. For example, in Figure 3, the horizontal expansion does not happen in slot 3, which can mean 2 level-hierarchy could be enough for solving the task. Ablation study on a different number of levels in the memory module can help to understand the importance of the hierarchical memory.
>
> We agree that the chosen tasks of Craft and Dial can essentially be solved by a 2-level hierarchy, and we will show the ablation study on the depths of the memory. However, we believe that the hierarchy is still important for several reasons:
>
> - From the methodology point of view, we would like to propose an architecture that is general enough, instead of only focusing on just 2-levels like many existing literature [1,2].
>
> - As we will show in the revision, even when we are training in a 2-level structure environment, our model is able to use a higher-level slot to build a multi-hierarchy structure. This shows the potential of applying our model to a "real" multi-level hierarchy environment.
>
> > In Figure 6, when the robot arm presses the number 4, the expected expansion position is high for multiple steps, which means the high-level task is likely to be changed over multiple steps. But this result seems not desirable. Is there any explanation why this still makes sense?
>
> Yes, it might not be desirable yet it still makes sense. The root cause for such behaviours is that we currently use the state to compute the expanding position. In Dial, the states between each time step has a less significant difference due to small time skip, and therefore the the expanding position will be similar across nearby time steps. When the state difference is more significance (such as in the discrete case of Craft, or continuous case with a larger time skip), the change of expanding position will be sharper. The observation mentioned might be due to the fact that the model stays close to the boundary state for multiple time steps.
>
> > In Figure 5, what is the difference between (a,b) and (c,d)?
>
> Sorry for the confusion. There is a typo in the captions. We will update the ablation results in the revision.
>
> About additional feedbacks:
> Thanks for your helpful suggestion. We will modify Figure 3 and Figure 4 accordingly in our next revision.
>
> [1] Kipf, Thomas, et al. "CompILE: Compositional imitation learning and execution." International Conference on Machine Learning. PMLR, 2019.
> [2] Shiarlis, Kyriacos, et al. "Taco: Learning task decomposition via temporal alignment for control." arXiv preprint arXiv:1803.01840 (2018).

---

> > ### Comment · AnonReviewer1 · 2020-11-20
> > **Thank you for your detailed response.**
> >
> > Thank you for your detailed response. This resolves most of my questions and concerns.
> >
> > However, the paper has not been updated yet. It would be great to see the revision with results on the Dial task and additional ablation studies.

---

> > > ### Author Response · Authors · 2020-11-23
> > > **Revision Updated**
> > >
> > > Dear Reviewer,
> > >
> > > We have updated the paper to address your concern with the ablation study and extra qualitative results on Kitchen. As the discussion phase is close to the end, we are looking forward to your further feedbacks

---

> ### Comment · Area_Chair1 · 2020-11-18
> **Author response**
>
> Dear AnonReviewer1,
>
> We are now entering the second discussion stage. Could you please check whether the authors have addressed your concerns and questions and potentially ask any further clarification questions?
>
> Thank you,
> Your Area Chair

---

> ### Author Response · Authors · 2020-11-23
> **Revision Updated**
>
> Dear Reviewer,
>
> We have updated our paper, and we are looking forward to your further feedbacks.

---

### Official Review · AnonReviewer2 · 2020-10-29
**Clear, easy to follow, missing comparisons.**

**Rating:** 6
**Confidence:** 4

**Review:**

Summary:
Paper introduces Ordered Memory Policy Network (OMPN) with an objective to learn sub-task decomposition and hierarchy from demonstrations in the context of Imitation Learning under unsupervised and weakly supervised settings. Authors approach the problem of uncovering the task substructure from the architecture design lens where the goal is to design architectures that have the right inductive biases leading to the discovery of task sub-structures. The proposed solution views sub-tasks as finite state machines represented as memory banks that are updated via top-down and bottom-up recurrences.

Strengths:
Paper is well polished and easy to follow. OMPN is shown to be effective in two domains. OMPN advantage in partially observable environments is also effectively demonstrated.

Discussion:
My reservations are with the evaluations.
- The choice of baselines also seems a bit narrow. Comparisons to related methods dealing with sub-task decomposition and organization are missing. For example, a comparison to the Relay Policy Learning[1] (which the paper claims to be the most related work) is missing. Another closely related method Learning Latent Plans from Play[2] (https://learning-from-play.github.io/) is also missing.
- Selected tasks have a very shallow task hierarchy and sparse task structure. The ceiling of the approach and limitations aren't clearly outlined and are less evident. This is also evident from the tasks considered in [1] and [2]. These methods have shown to be effective with play demonstrations in rich scenes such as kitchen and study-table scenes where are underlying task-structure is much more convoluted to uncover. Without similar comparisons, it's hard to evaluate the strength of OMPN. I'd strongly advise on including experiments under similar settings which will make the submission really strong.

---

> ### Author Response · Authors · 2020-11-16
> **Response Reviewer 2**
>
> Thank you for your feedback, and we are glad to find that you think the paper is well-polished. Here is the specific response to your questions in addition to the general response.
>
> > Selected tasks have a very shallow task hierarchy and sparse task structure. The ceiling of the approach and limitations aren't clearly outlined and are less evident. These methods have shown to be effective with play demonstrations in rich scenes such as kitchen and study-table scenes where are underlying task-structure is much more convoluted to uncover. Without similar comparisons, it's hard to evaluate the strength of OMPN. I'd strongly advise on including experiments under similar settings which will make the submission really strong.
>
> Thank you for your suggestion. We have looked into both datasets, and we already start running experiments on Kitchen. However, we find that there are no ground-truth task boundaries given, so we will update the qualitative results of Kitchen to show task decomposition we learned.
> For Study Desk, the dataset from Play-LMP is not released. The only information on the github is their webpage https://learning-from-play.github.io/.
>
>
> > The choice of baselines also seems a bit narrow. Comparisons to related methods dealing with sub-task decomposition and organization are missing. For example, a comparison to the Relay Policy Learning[1] (which the paper claims to be the most related work) is missing. Another closely related method Learning Latent Plans from Play[2] (https://learning-from-play.github.io/) is also missing.
>
> We strive to have a thourough and fair comparisions to the baseline methods. We thank the reviewer for pointing out the missing reference, and we will include them in the related work.  However, their paper might not be good baselines for our problem setting.
>
> Unlike our methods, the imitation learning (IL) phase of RPL and Play-LMP are not designed to produce trajectory segmentation, but to better support their following finetuning strategy. Although Play-LMP proposes some visual inspection on the learned latents but that is still far away from an executable skill-detection algorithm. Although we mention that RPL is the most related work in the paper, it is only within the context of HRL, but they are not directly comparable to us in terms of task decomposition.
>
> We thank the reviewer for the detailed feedbacks. Please let me know if you have any other questions.

---

> ### Comment · Area_Chair1 · 2020-11-18
> **Author response**
>
> Dear AnonReviewer2,
>
> We are now entering the second discussion stage. Could you please check whether the authors have addressed your concerns and questions and potentially ask any further clarification questions?
>
> Thank you,
> Your Area Chair

---

> ### Author Response · Authors · 2020-11-23
> **Revision Updated**
>
> Dear Reviewer,
>
> We have updated the paper, and we are looking forward to your further feedbacks.

---

### Author Response · Authors · 2020-11-14
**[Pre-revision] General Response to all Reviewers**

We thank all reviewers for the constructive insights. In addition to the specific response below, here we summarize our goals, address some common concerns and describe the changes planned to be included in the revision.
### Our Goal
Inspired by HAM [1], we study the inductive bias, so that the model can learn to expand multiple levels of subtask and return control back to the high level subtask, in a way described in Fig1. If we are able to achieve such a goal, then task decomposition would be trivial by just looking at the change of expanding positions. The task decomposition is a “by-product”， while our end goal is to design a model architecture which can operate with a hierarchy of subtasks. We plan to clarify this better in our revision.
### Our Achievement
1. We present the necessary architectural bias (sec2.1) and the regularization technique (sec2.2), while making sure that the model is end-to-end trainable and generally applicable to many environments. We demonstrate with visualization (Fig3, Fig6) that our design goal described above is achievable, with useful Information, like the number of subtasks, naturally emerges.

2. We further demonstrate that the task decomposition naturally follows with some easy post processing (e.g., top-K or thresholding). Our task decomposition results are comparable to (or better than) the baselines that require additional prior information.

### Common Concerns
1. We choose Craft and Dial as experimental environments mainly because they contain ground-truth structure annotations, published baselines, and well-defined quantitative metrics (e.g., f1 score, task alignment). These standards ensure that we can have quantitative proof that we can achieve our design goal and compare it with some baselines. Per the reviewer’s request, we conduct an additional experiment on the Kitchen dataset. Since there are no ground truth boundaries for quantitative evaluation, we will show the qualitative result in revision.

2. For the task decomposition, we agree with the reviewers that our detection algorithm relies on a specified number of subtasks K. However, this is consistent with the existing literature. Furthermore in our case, K seems to be readily available by visual inspection after training, which is something the baseline method cannot provide. To ensure fair comparisons, we set K to be the ground truth for both our methods and the baseline. In our revision, we will add the results when K is misspecified (i.e. not ground truth).

3. We agree with the reviewers that for the continuous case, currently we only show that an optimal threshold exists, but we did not provide an automatic way to find it. We now have an algorithm to select the threshold. We will show that its performance in the revision, even though the task decomposition is not our main goal here.

### Planned Changes
- We will add another ablation study to justify the design choice.
- We will do more hyperparameter analysis like varying the depths of the memory stack, and memory size.
- We will add an algorithm to automatically detect the threshold and demonstrate its effectiveness.
- We will show the detection algorithm results when K is not the ground truth value.
- We will address the typo mistakes as well as polish our figures and tables.
- We will add the performance of models on Dial.
- We will provide additional qualitative results on Kitchen

Please don’t hesitate to let us know for any additional comments on the paper or on the planned changes.

---

> ### Comment · AnonReviewer4 · 2020-11-18
> **The authors' response and proposed changes cover my concerns**
>
> I would like to thank the authors for their response and for sharing their planned changes.  I am satisfied with the proposed changes, though I believe the authors have not addressed one of my questions. In figure 4, they show some results for behaviour cloning on Craft World. I am wondering how the trained policy was obtained, or what kind of demonstrations/expert data they used to train on. I could not find in the paper. Could the authors provide some more information on that?

---

### Author Response · Authors · 2020-11-19
**Pre-revision Individual Response Updated**

Dear all reviewers,

Thank you for your detailed feedback. We have updated the individual response to each of your reviews under your thread. We will update the revision as well soon. Please let us know if you have further questions.

---

### Author Response · Authors · 2020-11-23
**General Response: Paper Revision Updated**


Dear Reviewers,
We would like to thank your thoughtful feedbacks. We are glad to see that reviewers generally appreciated the contributions of our paper -- the novelty of our architectural design to model the subtask hierarchy (R1, R4), writing clarity (R1,R2,R3), effective experimental results (R4), and the value of our method in future research (R3). Thank you for your patience, and we have provided an revision of our paper. We have updated our paper to include the following changes:

- The additional qualitative results on Kitchen is in Figure 5 and Appendix I. We show that the procedure on Dial could also give us reasonable task decompositon results for a more challenging setting.

- We update Figure 3 with icons to make it readable. We also show that our model could potentially learn a more than 2-level hierarchy by using the highest slot.

- In section 4.4, we perform the ablation study to justify our design choice of including bottom up/top down recurrences and including "done" loss as a regularization.

- The hyperparameter analysis with varying the depths of the memory stack, and memory size is done in Appendix H. We show that our results is robust to these hyper-parameters in most cases

- In Appendix B, we add an algorithm to autmatically detect the final threshold by computing an upper and lower value. In Table 7, We show that our automatic algorithm could produce competitive results to the best fixed threshold. Besides the final threshold we show the upper and lower value as well in Figure 4 & 5.

- We show the detection algorithm results when K is not the ground truth value in Table 5, 6 & 8.

- We update the behavior cloning curves on Dial with Figure 6.

- We provide details on demo generation in Appendix E.

---

### Decision · Program_Chairs · 2021-01-07
**Final Decision**

**Decision:**

Accept (Poster)

**Comment:**

This paper presents the Order-Memory Policy Network (OMPN), an architecture for modelling a hierarchy of sub-tasks and discovering task decompositions from demonstration data. Results are presented on a compositional grid-world task (Craft) and on a simulated robotics task (Dial).

The reviewers agree that the proposed method is novel and interesting, that the paper addresses an important problem, and that it is well-written. One main criticism by the reviewers, the lack of experimental evaluation of different hyperparameter choices, such as the depth of the memory stack and the expected number of subtasks, has to a large part already been addressed in the revision by the authors. The total number of hyperparameters that need to be tuned, however, is quite large and the authors are encouraged to revise their claim "Our central message is that OMPN is a general off-the-shelf model for task decompositions" in this light. The paper is borderline, and could clearly benefit from a revised, stronger presentation and more extensive experimental evaluation, but I am confident that the authors can use the time until the camera-ready version is due to address some of the remaining feedback by the reviewers, and hence I think that this paper can be accepted.

The authors are further encouraged to take the following additional reviewer feedback into account, which was brought up during the internal discussion period:
1) The complexity of the proposed method could be better justified by more thoroughly investigating the effectiveness of using a multi-level hierarchy (e.g., by running experiments on more complicated and hierarchical tasks with multiple branches).
2) Further strengthening down-stream performance evaluation, such as in imitation learning (in addition to the already presented behavioral cloning results) and/or reinforcement learning, would further strengthen the paper and demonstrate that the discovered decomposition is indeed useful.